# Effects of Diamond on Microstructure, Fracture Toughness, and Tribological Properties of TiO_2_-Diamond Composites

**DOI:** 10.3390/nano12213733

**Published:** 2022-10-24

**Authors:** Bing Liu, Zewen Zhuge, Song Zhao, Yitong Zou, Ke Tong, Lei Sun, Xiaoyu Wang, Zitai Liang, Baozhong Li, Tianye Jin, Junyun Chen, Zhisheng Zhao

**Affiliations:** 1Center for High Pressure Science (CHiPS), State Key Laboratory of Metastable Materials Science and Technology, Yanshan University, Qinhuangdao 066004, China; 2School of Mechanical Engineering, Yanshan University, Qinhuangdao 066004, China

**Keywords:** TiO_2_-diamond composite, hardness, fracture toughness, friction coefficient, high-pressure sintering

## Abstract

The reinforcements represented by graphene nanoplatelets, graphite, and carbon nanotubes have demonstrated the great potential of carbon materials as reinforcements to enhance the mechanical properties of TiO_2_. However, it is difficult to successfully prepare TiO_2_-diamond composites because diamond is highly susceptible to oxidation or graphitization at relatively high sintering temperatures. In this work, the TiO_2_-diamond composites were successfully prepared using high-pressure sintering. The effect of diamond on the phase composition, microstructure, mechanical properties, and tribological properties was systemically investigated. Diamond can improve fracture toughness by the crack deflection mechanism. Furthermore, the addition of diamond can also significantly reduce the friction coefficient. The composite composed of 10 wt.% diamond exhibits optimum mechanical and tribological properties, with a hardness of 14.5 GPa, bending strength of 205.2 MPa, fracture toughness of 3.5 MPa∙m^1/2^, and a friction coefficient of 0.3. These results enlarge the family of titania-based composites and provide a feasible approach for the preparation of TiO_2_-diamond composites.

## 1. Introduction

Titania (TiO_2_) has been extensively applied in solar cells [1], photoelectric devices [2], chemical sensors [3], and biomaterials [4] due to its excellent properties, such as high chemical and thermal stability, good biocompatibility, and low production cost [5,6,7,8,9]. In these applications, not only a balance of strength and toughness, but also good tribological properties are essential for long-term service.

In fact, due to the low mechanical properties of pure TiO_2_ (especially fracture toughness) and the tendency of large grain growth, titania-based composites are the main form of the practical application of titania [10,11,12]. For example, Miao et al. [10] prepared titania-yttria-stabilized tetragonal zirconia (TiO_2_/15 vol%-Y-TZP) composites, which have a hardness value of 9.8 GPa, bending strength of 160 MPa, and fracture toughness of 3.8 MPa·m^1/2^. Nadaraia et al. [13] produced titania–graphene nanoplatelet (TiO_2_/GNP) composites using the spark plasma sintering (SPS) technique. The hardness of TiO_2_/GNP composites is in the range of 9.2~10.6 GPa, and the fracture toughness is in the range of 2.3~9.6 MPa·m^1/2^, indicating the great potential of carbon materials as a reinforcement to enhance the mechanical properties of TiO_2_. In addition, graphite and carbon nanotubes as reinforcements have also been reported in the literature [14,15]. However, there are few reports on TiO_2_-diamond composites. One of the key obstacles is that diamond, especially nanoscale diamond, is highly susceptible to oxidation or graphitization at relatively high sintering temperatures [16,17,18]. To our knowledge, the only report is from Hanada et al., who successfully sintered TiO_2_-diamond composites using SPS [19]. TiO_2_ existed in the forms of rutile phase, or mixed rutile and anatase phase in Hanada’s work. Moreover, the decrease of hardness of TiO_2_-diamond composites with increasing volume fractions of diamond was reported. In addition, it was found that the friction coefficient first decreased, and then increased as the diamond content increased. The addition of diamond leads to a 10% reduction of the friction coefficient, with the highest hardness of about 9 GPa.

High pressure (several GPa) can effectively reduce the sintering temperature while ensuring consolidation [20,21,22,23]. It has recently been reported that TiO_2_ nanoceramics can be sintered at a low temperature, around 500 °C and 6 GPa, and the hardness of the synthesized sample can reach at least 9.8 GPa [24]. Inspired by these studies, the current work prepared TiO_2_-diamond composites using high-pressure sintering. The changes in phase composition, microstructure, mechanical properties, and tribological properties with diamond content were systemically investigated. The optimal sample with a good balance of mechanical properties, including hardness, bending strength, fracture toughness, and tribological properties, was obtained. These results provide a new approach for the preparation of TiO_2_-diamond composites.

## 2. Materials and Methods

### 2.1. Sample Synthesis

Commercially available anatase-type TiO_2_ powders (purity: 99.6%, particle size: ~100 nm, Alfa Aesar, Haverhill, MA, USA) and diamond powders (purity: 99.9%, particle size: ~800 nm, Alfa Aesar) were used as raw materials. The TiO_2_-diamond composites were prepared with six different diamond contents, viz. 0, 10, 20, 30, 40, and 50 wt.%. These samples were named T-0%D, T-10%D, T-20%D, T-30%D, T-40%D, and T-50%D, respectively. Powder mixtures were made through ball-milling in ethanol for 1.5 h. After ball-milling and drying, the powder mixtures were treated at 200 °C and high vacuum (3.0 × 10^−5^ Pa) for 2 h to remove volatile impurities in the powders. High-pressure experiments at pressures of 6 GPa and temperatures of 450 °C were performed in a DS 6 × 8 MN cubic press machine (Guilin Guiye Machinery Co., Ltd., Guilin, China), that is, a cubic anvil apparatus, by using a standard sample assembly consisting of a pyrophyllite and graphite heater. The temperature was measured with type-C thermocouples, and the pressure was estimated from previously determined calibration curves. During the synthesis, pressure increased to 6 GPa in one minute; the sample was then heated at a rate of 50 °C per minute to the target temperature. In all the experiments, the sample was maintained under the target pressure and temperature for 1 h. After that, the electric power supply was then turned off and the sample was quenched to room temperature, followed by a pressure release at a rate of 0.3 GPa per minute to atmospheric conditions. The recovered samples were 8–10 mm in diameter and height. All test samples were polished using a multi-stage diamond abrasive paste for further characterization.

### 2.2. Microstructure Characterization

Phase analysis was performed by X-ray diffraction (XRD, Smartlab Rigaku, Japan) with a Cu Kα radiation source (λ = 1.5406 Å). The XRD data were refined by TOPAS 4.2 software. The morphologies of raw materials and synthesized samples were characterized using scanning electron microscopy (SEM, Verios G4 UC, Thermo Fisher Scientific, Waltham, MA, USA). Microstructures of the samples were characterized using transmission electron microscopy (TEM, ARM200F, JEOL, Tokyo, Japan) with an accelerating voltage of 200 kV. The samples for TEM observation were prepared using a Ga-focused ion beam (Scios, FEI, Hillsboro, OR, USA), milling with an accelerating voltage of 30 kV. The pieces with a size of ~10 μm × 10 μm × 2 μm were first precut from the bulk samples by using a current of 30 nA. Ion beam currents from 7, 5, 3, 1, 0.5, to 0.1 nA were used in sequence to mill the pieces to electron-transparent slices with thicknesses of less than 100 nm. Subsequently, ion cleaning for approximately 10 min was applied to each side of the slice under a voltage of 5 kV and a current of 16 pA to minimize the knockout damage on the slices.

### 2.3. Mechanical Properties and Tribological Properties Test

The three-point bending tests were performed on the mechanical properties testing system (MTII/Fullman SEM tester 2000, USA) at room temperature. The three-point bending specimen was a cuboid with a size of 0.6 mm × 1.2 mm × 4.5 mm, the test span was 3 mm, and the crosshead speed was 0.05 mm/min. The nanoindentation tests were performed by using a Nano Indenter G200 (KLA-Tencor, Milpitas, CA, USA) with a Berkovich diamond tip at an applied load of 200 g, to measure the hardness and elastic modulus of the synthesized samples. The applied standard loading time to peak load was 15 s, the peak holding time was taken as 10 s, and the unloading time was 15 s. The hardness was calculated by the Oliver and Pharr method [25]. The fracture toughness of the synthesized samples was tested using a Vickers indenter (KB 5 BVZ) at an applied load of 800 g. The fracture toughness (*K*_IC_) measurements were determined as *K*_IC_ = 0.203(*c*/*a*)^−3/2^·*H*_v_·*a*^1/2^ [11,24], where *c* (μm) is the average length of the radial cracks measured from the indentation center, *a* (μm) is one half of the average length for the two indent diagonals, and *H*_v_ is the hardness. 

The tribological properties measurements (Anton Paar, Graz, Austria) were performed under ambient atmosphere conditions (50% relative humidity) and used silicon nitride balls (diameter 3 mm) as the counterpart material. The applied normal load was 1 N while the sliding stroke and frequency were 3 mm and 2 Hz, respectively. The total swipe time was 20 min.

## 3. Results

The SEM images of the pristine diamond and anatase-type TiO_2_ powder are shown in Figure 1a,b. The diamond particles have a polyhedron shape, and the TiO_2_ particles are approximately spherical. The average particle sizes of the diamond and TiO_2_ particles are around ~800 nm and ~100 nm, respectively. Figure 1c shows the XRD patterns of the raw materials and powder mixtures. The XRD patterns of the powder mixtures consist only of diffraction peaks corresponding to the diamond and anatase-type TiO_2_, indicating no impurities were introduced during the powder mixing process. In addition, the intensity of the diffraction peak (2*θ* = 43.9°) corresponding to the (111) plane of diamond in the samples from T-0%D to T-50%D increased gradually; this was consistent with the gradual increase in the diamond content of the powder mixtures.

The XRD patterns of synthesized samples are given in Figure 2a. The diffraction peak corresponding to diamond can be seen in the figure, and no diffraction peaks belonging to graphite or titanium carbide were detected. These results indicate that the current synthesizing conditions avoids the graphitization of diamond, as well as the reaction between TiO_2_ and diamond. In addition, the diffraction peaks belonging to the rutile phase and columbite phase can also be detected, due to the phase transformation of TiO_2_, which is consistent with the previous research results [24]. It is worth noting that there are slight differences in the phase composition of TiO_2_ of the samples for different diamond additions. With the increase in diamond additions, the intensity of the diffraction peaks of the columbite phase gradually decreases, and the intensity of diffraction peaks corresponding to the rutile phase gradually increases. An incompletely transformed anatase phase was observed for the composite containing 50 wt.% diamond. Furthermore, the quantitative results of phase composition obtained by Rietveld refinement are shown in Figure 2b. With the increase in diamond additions, the content of the columbite phase rapidly decreases from nearly 87.3 wt.% in the T-0%D sample to 21.3 wt.% in the T-50%D sample. The content of the rutile phase in the composites increases with diamond content and reaches 23.7 wt.% when the diamond content is 50 wt.%. These results indicate, on the one hand, diamond hinders the phase transformation from anatase to columbite, and on the other hand, promotes the phase transformation from anatase to rutile. As the diamond content increases, the effect of hindering phase transformation becomes more pronounced; thus, when the content of diamond in the composite is 50 wt.%, the remaining anatase phase reaches 5.1 wt.%.

Furthermore, the distribution of diamond particles in the TiO_2_ ceramic matrix was investigated using SEM, and Figure 3 shows the composite surface after polishing with multi-stage diamond abrasive paste. In this figure, the gray contrast corresponds to the TiO_2_ matrix, and the bright contrast represents the diamond. It can be clearly seen that diamond particles are uniformly distributed in the synthesized samples when the diamond content is less than or equal to 30 wt.%, as shown in Figure 3b–d. The diamond particles were tightly embedded in the TiO_2_ nanoceramics, and there was no trace left due to diamond peeling during the grinding and polishing process. This result provides evidence of a strong bond between the diamond and TiO_2_. When the diamond content is larger than 30 wt.%, agglomeration phenomena of the diamond can be observed in synthesized samples.

The detailed microstructures of the TiO_2_-diamond composites with the different diamond additions were further investigated by TEM analysis. Figure 4a is a representative high-angle annular dark field (HAADF) image of the T-10%D sample. In this figure, the gray contrast corresponds to the TiO_2_ matrix, and the dark contrast represents the diamond. Submicron-scale diamond particles are tightly wrapped around the TiO_2_ matrix and no obvious pores or cracks have been found. Furthermore, the corresponding energy-dispersive X-ray spectroscopy (EDS) maps confirm this result. When the diamond content is more than 20 wt.%, microcracks that initiate from the tips of diamond particles and distribute in the TiO_2_ matrix can be observed (Figure 4c–e). The appearance of microcracks may be caused by the stress concentration during high-pressure sintering due to the difference in the physical properties of diamond and TiO_2_.

The interface binding of diamond particles and TiO_2_ matrix is further investigated using a TEM bright field (BF) image with high magnification and high-resolution TEM (HRTEM). The results are given in Figure 5. From Figure 5a, it can be found that the diamond particles are wrapped tightly by the TiO_2_ matrix, and there are no voids and microcracks along the TiO_2_/diamond interface. A similar phenomenon can also be observed in Figure 5b. Taking the interface of diamond and columbite-type TiO_2_ as an example, Figure 5b displays a clear interface without second phases, and an amorphous layer. These results indicate exactly a firm interface binding between diamond particles and the TiO_2_ matrix. It should be noted that several microcracks in Figure 4d,e deflected along the interface between diamond particles and TiO_2_ matrix, indicating a weak binding in the T-40%D and the T-50%D samples.

The mechanical properties of TiO_2_-diamond composites were further investigated. Using the nanoindentation test, the hardness of the synthesized samples were examined, and the results are given in Figure 6. With the increase of diamond content, the hardness decreases. Similar results have been reported in previous research [19]. Since the columbite-type TiO_2_ has a higher hardness than the rutile phase and anatase phase [24], it is reasonable to believe that the decrease in columbite content with increasing diamond is responsible for the decrease in hardness. For the T-0%D sample, the hardness is 15.2 GPa. When the diamond content is 10 wt.%, a high hardness of 14.5 GPa is present. In addition, it should be noted that this hardness (14.5 GPa) is higher than the hardness of titania-based composites reported in previous literature (9.8 GPa [10], 8.7 GPa [12], 9.2~10.6 GPa [13], 9.0 GPa [19]), although the hardness decreases slightly with the increase of diamond content. Figure 7 shows the bending strength of synthesized samples. Similar to hardness results, the bending strength is basically invariable when the diamond content is less than or equal to 20 wt.%, and abruptly decreases when the diamond content is more than 20 wt.%. The abrupt decrease may be related to the aggregation of diamond particles (Figure 3e,f) and the existence of microcracks (Figure 4b–e). Notedly, the bending strength of the T-10%D is up to 205.2 MPa, which is also much higher than the bending strength of titania-based composites reported in previous literature (160 MPa [10]).

The effect of diamond on fracture toughness was also investigated. In this work, the fracture toughness was examined using a Vickers indenter. Because of the cliff-like decline of hardness and bending strength in the T-30%D, T-40%D, and T-50%D samples, this work focuses on the T-0%D, T-10%D, and T-20%D samples to examine the fracture toughness and tribological properties. The fracture toughness of the T-10%D and T-20%D samples are 3.5 MPa∙m^1/2^ and 3.9 MPa∙m^1/2^, respectively, which is about twice that of T-0%D (1.8 MPa∙m^1/2^). Obviously, the diamond toughens the TiO_2_-diamond composites. The operative toughening mechanism can be speculated from the SEM image of the fracture surface. Figure 8 displays a representative SEM image of Vickers indentation of the T-10%D sample. In this figure, the radial cracks can be clearly observed. From the magnified image (Figure 8b), it can be found that the tip of cracks would deflect when they encountered the diamond particles, indicating diamond can improve the fracture toughness by the crack deflection mechanism.

Figure 9 shows the friction coefficient curves of TiO_2_-diamond composites as a function of time. During the first 100 s, the friction coefficient increases with the increase of time, and then the friction coefficient remains basically constant or fluctuates within a certain range. In the measured conditions, the friction coefficient of the T-0%D sample is about 0.9, while the friction coefficients of the T-10%D and T-20%D samples are about 0.3 and 0.6, respectively. In both cases, the friction coefficients tend to decrease due to the addition of diamond, which is consistent with the previous research [19,26,27,28,29,30]. For the T-20%D samples, an obvious fluctuation is observed in the friction coefficient curves. This phenomenon may be related to the distribution of diamond particles in the T-20%D samples. Notedly, the addition of diamond can reduce the friction coefficient by up to 66.7%, possibly due to the low friction coefficient of pure diamond (about 0.1 [31]) and the rolling effect of diamond particles suggested in previous studies [26,27,28,30].

## 4. Conclusions

In this work, a series of TiO_2_-diamond composites were prepared using high-pressure sintering. The effect of diamond on the phase composition, microstructure, mechanical properties, and tribological properties was systemically investigated. The main conclusions are listed as follows.

(1)Diamond can promote the phase transformation from anatase to rutile and hinder the phase transformation from anatase to columbite.(2)Hardness and bending strength decrease with the increase of diamond content.(3)Diamond can improve the fracture toughness of TiO_2_ by the crack deflection mechanism.(4)The addition of diamond reduces the friction coefficient by up to 66.7%.(5)To enhance the mechanical properties of the TiO_2_ matrix, the diamond content needs to be limited in range, i.e., less than or equal to 20 wt.%. The composite composed of 10 wt.% diamond exhibits optimum mechanical and tribological properties, with a hardness of 14.5 GPa, bending strength of 205.2 MPa, fracture toughness of 3.5 MPa∙m^1/2^, and a friction coefficient of 0.3.

## Figures and Tables

**Figure 1 nanomaterials-12-03733-f001:**
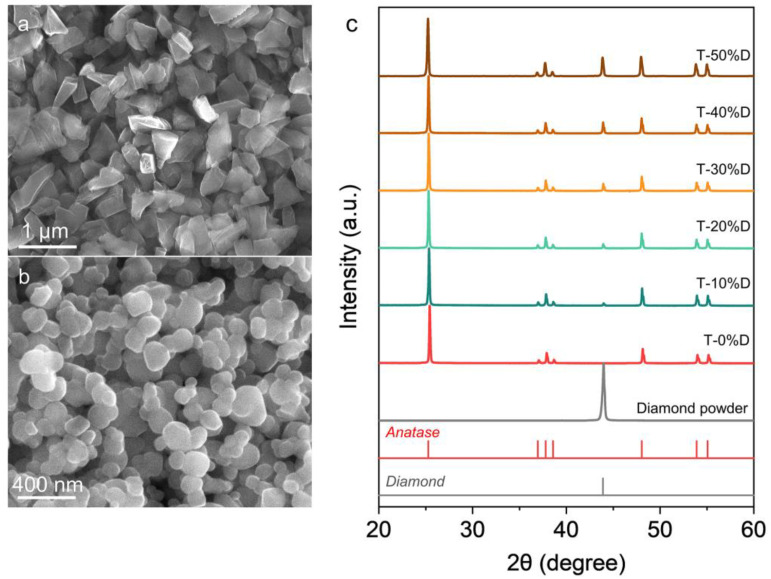
The SEM images of the raw materials: (**a**) diamond, (**b**) anatase-type TiO_2_. (**c**) XRD patterns of the raw materials and powder mixtures.

**Figure 2 nanomaterials-12-03733-f002:**
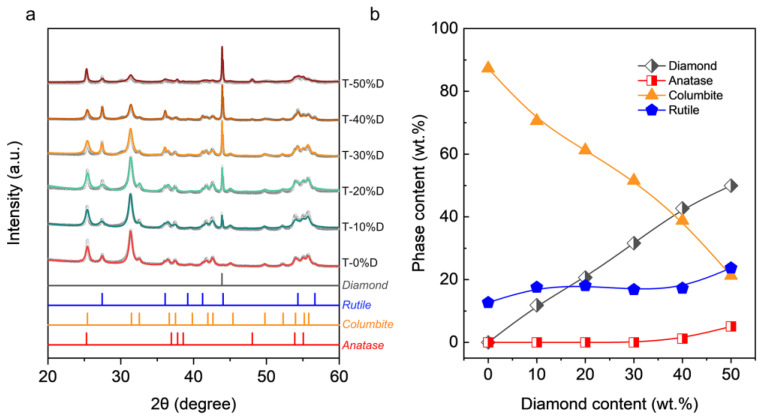
(**a**) Experimental and Rietveld refined XRD patterns of the recovered samples of TiO_2_-diamond composites sintered at 6 GPa and 450 °C. (**b**) Phase content variation as a function of diamond content.

**Figure 3 nanomaterials-12-03733-f003:**
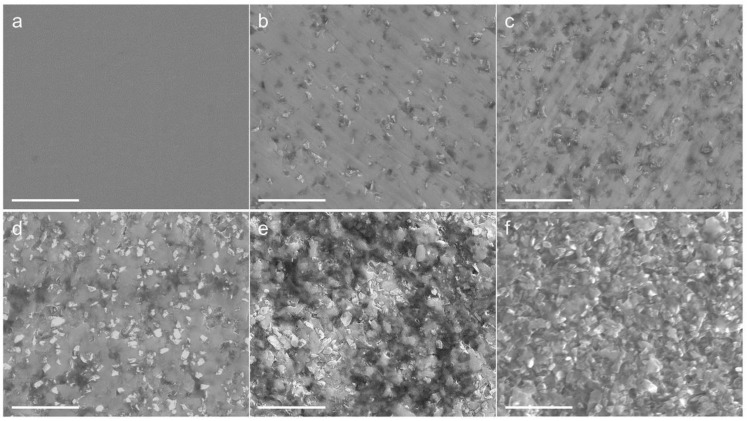
SEM micrographs showing the microstructure of the TiO_2_-diamond composites. (**a**) T-0%D, (**b**) T-10%D, (**c**) T-20%D, (**d**) T-30%D, (**e**) T-40%D, (**f**) T-50%D. The scale bars are 5 μm.

**Figure 4 nanomaterials-12-03733-f004:**
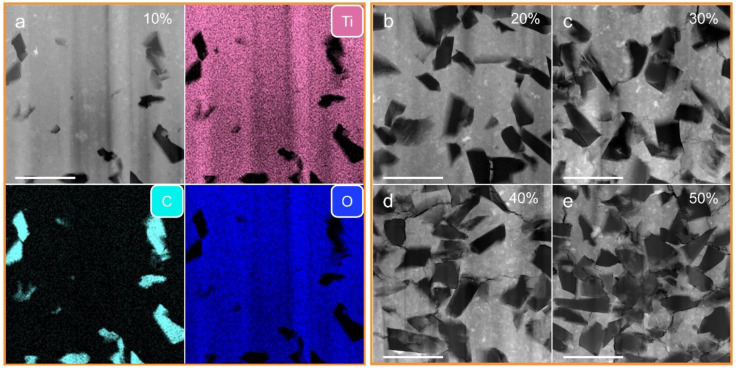
HAADF images of the TiO_2_-diamond composites. (**a**) T-10%D, (**b**) T-20%D, (**c**) T-30%D, (**d**) T-40%D, (**e**) T-50%D. The corresponding EDS maps of the T-10%D sample are also given in (**a**). The scale bars are 1 μm.

**Figure 5 nanomaterials-12-03733-f005:**
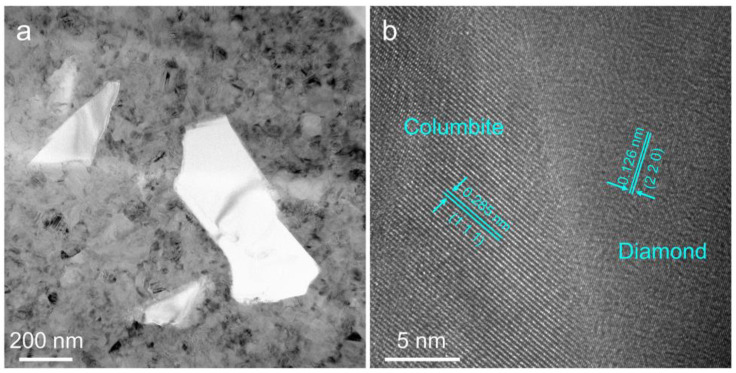
The TEM micrographs showing the interface binding of diamond particles and TiO_2_ matrix in the T-10%D sample. (**a**) BF image, (**b**) HRTEM image.

**Figure 6 nanomaterials-12-03733-f006:**
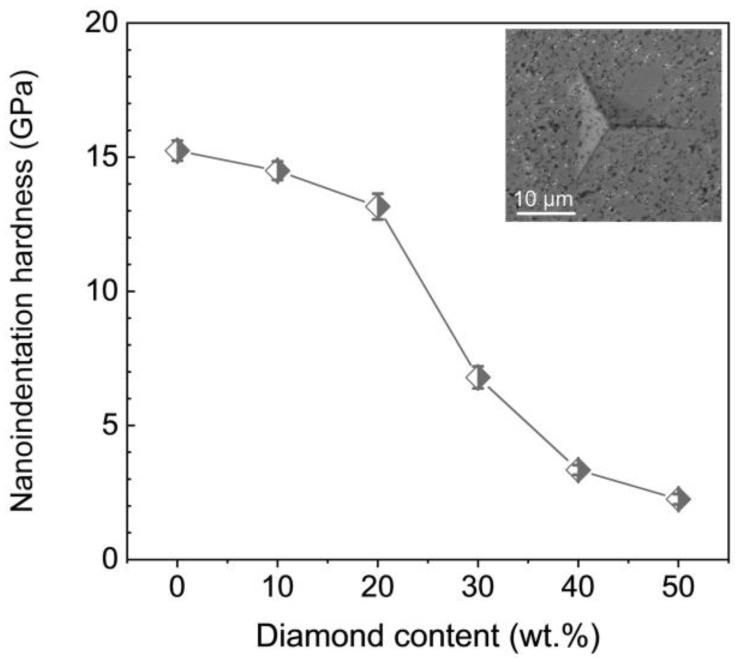
The nanoindentation hardness as a function with diamond content. Inset is the indentation photograph of the T-10%D sample under a load of 200 g.

**Figure 7 nanomaterials-12-03733-f007:**
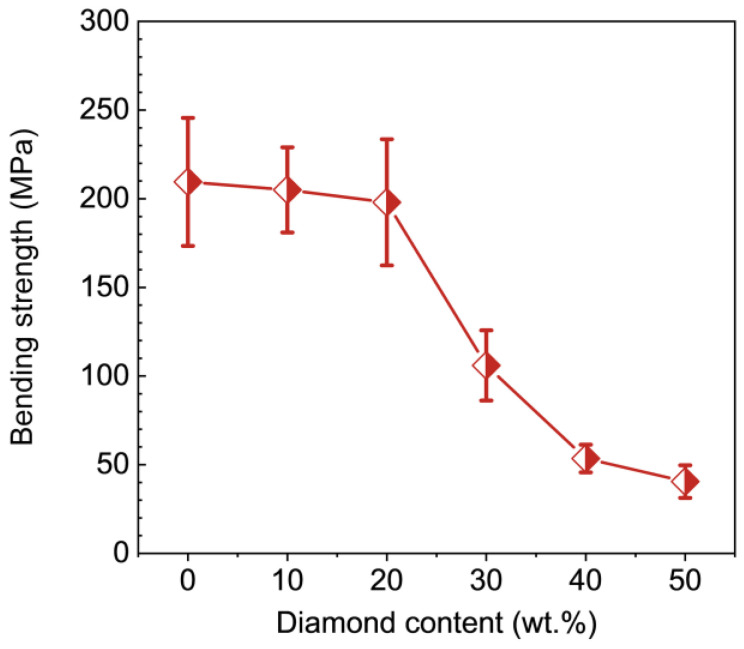
Bending strength of the TiO_2_-diamond composites as a function of the diamond content.

**Figure 8 nanomaterials-12-03733-f008:**
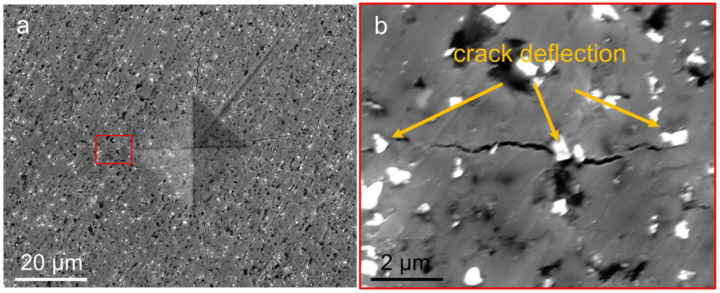
(**a**) SEM micrographs of the fracture surface of the T-10%D sample. (**b**) Magnified SEM image corresponding to the red-boxed area in (**a**).

**Figure 9 nanomaterials-12-03733-f009:**
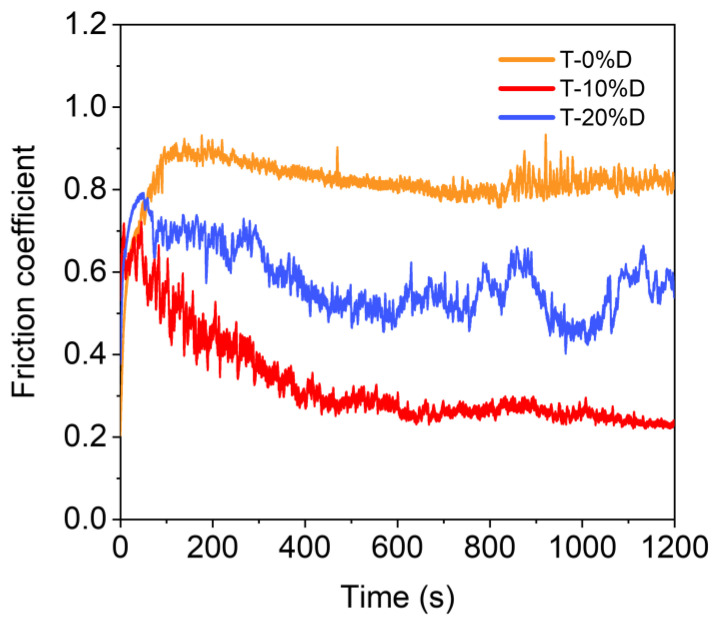
The friction coefficient of the TiO_2_-diamond composites as a function of time.

## Data Availability

The data that support the findings of this study are available from the corresponding writer upon reasonable demand.

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
