# Peer review of "Effects of Diamond on Microstructure, Fracture Toughness, and Tribological Properties of TiO2-Diamond Composites"

_nanomaterials, 2022, doi:10.3390/nano12213733_

Round 1

Reviewer 1 Report

In this article, the authors prepared TiO2/diamond composite at low temperature sintering with varied ratio of diamond and examined the mechanical and tribological properties. Not any specific hypotheses were made but a high-pressure sintering method was demonstrated. The final conclusions are sound in scientific points of view and properly backed by the results. Nevertheless, I got two concerns about this study: 1) Columbite is a metastable polymorph of TiO2 that forms at high pressure. The possible re-transformation to a low temperature stable phase and its possible effect on the mechanical properties should be also discussed. 2) Considering the moderate improvement in the examined properties as compared to other type of reinforcing additives, the high price of nanodiamonds made little sense of using such a composite in real applications. Some more minor comments are as follows:

- - The ratio of the nanodiamonds should be given in volume ratio instead of weigth%.

  - More details is needed about sample synthesis, including e.g. the dimensions of the sintered samples, the heating method. Was the sample under pressure throughout the heating, if so, how it could be heated. The sample was quenched after heat treatment but in what way?

   - It is quite interesting that the hardness of the composite decreases with increasing the ratio of the hardest filler material. This phenomenon calls for some comment in the article.

Reviewer 2 Report

The study deals with the preparation and investigation of the properties of the TiO2 -diamond composites. The results of the study are clearly written, conclusions are relevant and the study presents an interest for scientific society.  

  There are only minor questions, comments and suggestions.

 1. The main concern is a lack of discussion of the binding forces issue between the diamond and TiO2. It was found that between 10 and 40wt. % diamond significant changes occur in properties of composites and it was explained by agglomeration of diamond.

 - The Introduction should be extended with analysis how the size, concentration and other properties of diamond and matrix material influence on the properties of the diamond-TiO2 or some similar compounds.

 - As well as the Conclusions should contain a part regarding to this finding, i.e. existence of the limit in concentration of 10-20wt. % diamond (with particular size of diamond or maybe other properties) leading to the significant changes in properties of composite.

2. Authors write  that diamond, especially nanoscale diamond, is highly susceptible to oxidation or graphitization at relatively high sintering temperatures [16-18].

In the body of the text, the main reason of the degradation of the properties is diamond agglomeration of diamond. Please discuss in more details the possible reasons for the properties degradation between 10-40 wt. % diamond?

3. The results show firm interface binding between the diamond and TiO2. Diamond is a very inert material, what kind of the binding forces between the matrix and filler lead to a such conclusion? Could be that high compressive stress is the “adhesive force”? Discussion of this topic would be also desirable.   
